# Spatial Pattern and Fairness Measurement of Educational Resources in Primary and Middle Schools: A Case Study of Chengdu–Chongqing Economic Circle

**DOI:** 10.3390/ijerph191710840

**Published:** 2022-08-31

**Authors:** Wei Lu, Yuechen Li, Rongkun Zhao, Bo He, Zihua Qian

**Affiliations:** 1Chongqing Jinfo Mountain National Field Scientific Observation and Research Station for Karst Ecosystem, Chongqing Engineering Research Center for Remote Sensing Big Data Application, School of Geographical Sciences, Southwest University, Chongqing 400715, China; 2Key Laboratory of Monitoring, Evaluation and Early Warning of Territorial Spatial Planning Implementation, Ministry of Natural Resources, Chongqing 401147, China; 3Chongqing Planning & Design Institute, Chongqing 401147, China

**Keywords:** point of interest, educational resources of primary and middle schools, spatial pattern, educational equality, Chengdu–Chongqing economic circle

## Abstract

China’s education has developed rapidly in recent years, but the issue of educational equality still exists. Currently, there are few studies on educational resources, and their spatial pattern and fairness remain unclear. Thus, this study selected the point of interest data and spatial analysis methods to depict the spatial pattern of educational resources (containing the number of teachers, students, facilities, etc.). Then, we evaluated the equity of educational resources (including the number of schools and school teachers) in terms of geographic and population distribution by combining statistical yearbook data with two indices (the index of dissimilarity and agglomeration degree) to promote healthy urban development. The results show the following. (1) Educational resources have a multicenter spatial structure of “dual cores and multiple sub-centers”. The Moran index reflects a weak positive spatial correlation between educational resources. (2) The index of dissimilarity is between 0.02 and 0.21, which shows that the allocation of resources is relatively balanced. Regarding internal units, obvious differences exist in the agglomeration degree and equilibrium of educational resources.

## 1. Introduction

Education, as an important element of human development, is conducive to leading us to form the correct world view, outlook on life, and values. It also helps reduce poverty, socio-economic gaps, crime rates, etc. [1,2]. In 2015, ensuring inclusive and equitable quality education was included as one of the United Nation’s Sustainable Development Goals. However, the uneven distribution of educational resources remains a huge obstacle to achieving this goal [3,4,5]. As the basic unit that carries educational resources, schools have an important impact on the optimization of the settlement networks and the improvement of house prices, etc. [6,7]. Thus, it is vital to study the spatial distribution of schools. In China, the education system can be divided into pre-primary education (for children ranging from the age of 3 to 6 years), basic education (for children ranging from the age of 6 to 12 years), secondary education (for children ranging from the age of 12 to 18 years), and higher education (for adults ranging from the age of 18 to 27 years). Primary and middle schools (PMSs), as important institutions providing basic and secondary education, are major factors affecting the national development level and international competitiveness. Some studies have shown that the spatial distribution of PMSs is uneven, which is closely related to traffic space and living space [8,9]. Hence, it is valuable to focus on the spatial pattern and equity of educational resources in PMSs.

Research on equity in education has been extensively conducted around the world, mainly focusing on gender, age, ethnicity, disability, resources, and socio-economic status [10,11,12,13,14,15]. In traditional studies, the standard deviation of years of schooling among students was often used as an indicator for measuring equity in educational resources [16,17]. Nevertheless, compared to the standard deviation, the Gini coefficient is a better measure [18]. Some scholars attempted to evaluate the equity of educational resources in different regions through the Gini coefficient [1,3,19,20]. In these studies, the Gini coefficient declined significantly in most regions (i.e., educational inequality has decreased), but disparities in educational development between regions remain large. A recent study, based on an output-based approach, measured equity of education resource (containing teacher quality, school physical environment, and school instructional environment) allocation in Brazil, United States, Pakistan, and Peru [21]. The results indicated that four regions have a substantial mismatch of resources and needs, especially in Brazil. Notably, the statistics used in the above studies may be incomplete or difficult to obtain. However, analyzing educational resources from a spatial perspective (i.e., using spatial data) can be an effective approach to address this issue. Early research has focused mainly on the connection between the spatial layout of educational resources and the structure of macro-political and economic [22,23]. Subsequently, Hones and Ryba [24] proposed a basic research framework for educational resources (i.e., the “space-mechanism effect”) to analyze the spatial pattern of educational resources from multiple angles and methods, and then explored the spatial mechanism and effects behind this phenomenon. With the maturity of theory and technology, scholars gradually have turned to measure the spatial pattern and equilibrium of educational resources from different scales [25,26,27]. In addition, scholars also have associated the spatial pattern of educational resources with the process of urbanization [28,29,30] and regional planning [31,32,33] to reveal the mechanism of interaction between them. Recently, accessibility has gradually become a widely used method to measure the equity of educational resources [34,35,36,37,38]. In these studies, a common phenomenon was revealed, i.e., the inequitable distribution of educational resources accessibility. In addition, geographical information system and spatial analysis are effective ways to reveal the spatial pattern of educational resources and promote equity in education. For example, Yuan et al. [39] used statistical maps to show the spatial pattern of China’s high-quality elementary educational resources and then mined its formation mechanism through multiple linear regression. The results identified obvious differences between the east and the west terms of high-quality elementary education, and the formation of high-quality elementary education was highly correlated with the level of economic and teacher. Luo et al. [40] explored the spatial distribution features and affecting factors of shadow education institutions in Lanzhou by methods of the geographical detectors, etc. They found that educational resources were sparse in the west and dense in the east. Surrounding facilities and traffic accessibility were important factors affecting the distribution of resources. Wang et al. [41] evaluated the equilibrium of spatial of public educational resources in Beijing through first-degree analysis and benefit end-result analysis. They found that the benefit areas of public education expenditure present the spatial characteristics of circle distribution. Furthermore, a recent study revealed the spatial pattern of educational resources in urban fringe areas through the Geo-information Tupu and density-field-based hotspot detector [42]. This has important implications for exploring differences in the equity of educational resources between regions (i.e., urban areas, urban fringe areas, and rural areas).

In conclusion, scholars have explored the spatial pattern and equity of educational resources from different perspectives, but these studies have limitations. Traditional methods for measuring the fairness of educational resources mainly include the Lorenz curve, coefficient of variation, and Gini coefficient [20]. However, these methods only analyze the fairness of educational resources from the macro scale of supply and demand, which is difficult to effectively reflect the real situation. For accessibility, some simple modeling ignores the students’ choice of school and the complex behavior of school travel [43,44,45]. For geographical information system and spatial analysis, a quantitative approach is lacking to reveal the reasons behind the uneven spatial distribution [8]. To address these issues, we applied the index of dissimilarity (ID) and agglomeration degree to measure the equity of educational resources. Moreover, we employed the kernel density estimation (KDE), standard deviation ellipse (SDE), and spatial autocorrelation analysis to reflect the spatial pattern of educational resources. This study can provide decision-making support for adjustment of educational resources and future planning.

## 2. Study Area and Materials

In January 2020, the concept of the Chengdu–Chongqing economic circle was first proposed. It is an urbanization area with the highest level of development and significant development potential in western China, and an important part of the implementation of the Yangtze River Economic Belt and the One Belt One Road strategy [46]. Recently, He et al. [47] indicated that the layout and structure of educational resources in the Chengdu–Chongqing economic circle had existing problems (the phenomenon of “collapse in the central part”). Consequently, exploring the spatial pattern and equity of educational resources in the region is important for sustainable urban development. Figure 1 shows the scope of the study area. To facilitate the overall analysis and ensure regional connectivity, Shizhu Tujia Autonomous County and Pengshui Miao Tujia Autonomous County were included in the research scope. Meanwhile, nine districts of Chongqing (Shapingba, Jiulongpo, Beibei, Banan, Dadukou, Yubei, Yuzhong, Nan’an, Jiangbei) were merged into the main urban area of Chongqing (MUAC), so this study involved 38 basic units.

The data selected in this study included vector data of administrative divisions, point of interest (POI) data, and statistical yearbook data. The vector data of administrative divisions came from the National Geographic Information Resource Directory Service System (https://www.webmap.cn, accessed on 11 October 2020). We obtained POI data by using the API of Amap in December 2020 and obtained 12,221 valid data points after subsequent cleaning (including eliminating repeated values and outliers) (https://lbs.amap.com/api/webservice/summary, accessed on 26 December 2020). According to statistics, the Chengdu–Chongqing economic circle had 7696 primary schools and 4525 middle schools (including junior high school and senior high school). Figure 2 shows the number of PMSs in each study unit. Statistical yearbook data came from China’s economic and social big data research platform (https://data.cnki.net, accessed on 18 November 2020). It specifically included the number of full-time teachers, students, and schools in 38 research units of the Chengdu–Chongqing economic circle from 2010 to 2019.

## 3. Methodology

In this study, we measured the spatial pattern and fairness of educational resources. Firstly, as schools are the basic unit of educational resource allocation, we explored its spatial pattern based on POI data (coordinate points of PMSs). Of these, the methods used were SDE, KDE, and spatial autocorrelation analysis (Section 3.1, Section 3.2 and Section 3.3). Secondly, we investigated the equity of educational resources in terms of population distribution and geographic distribution based on statistical yearbook data. Of these, the methods used were ID and agglomeration degree (Section 3.4 and Section 3.5).

### 3.1. Standard Deviation Ellipse

SDE is widely used in the quantitative description of spatial features of ground objects. Its center, long axis, short axis, and azimuth parameters can reflect the central position, direction trend, and dispersion degree of geographic elements [48]. Therefore, we selected the SDE to reveal the spatial form and differences of PMSs in the study area. The calculations are shown in Equations (1) and (2):(1)$$SDE_{x}=\sqrt{\frac{\sum_{i=1}^{m}{\left(x_{i}-\bar{X}\right)}^{2}}{m}},$$
(2)$$SDE_{y}=\sqrt{\frac{\sum_{i=1}^{m}\left(y_{i}-\bar{Y}\right)}{m}},$$
where $SDE_{x}$ and $SDE_{y}$ represent the center of the ellipse, $x_{i}$ and $y_{i}$ represent the geographic coordinates of school i, $\left\{\bar{X},\bar{Y}\right\}$ represents the average center of schools, and m is the total number of schools.

### 3.2. Kernel Density Estimation

KDE is based on the “First Law of Geography”, that is, near things are more related to each other. In this study, Silverman’s quadratic kernel function was selected to reckon the density contribution value of the sample points to the grid cell center within the specified radius to characterize the relative agglomeration degree of PMSs in continuous areas of the Chengdu–Chongqing economic circle [49]. The calculation is shown in Equation (3):(3)$$\hat{f}\left(x,y\right)=\frac{3}{\pi nr^{2}}\sum_{i=1}^{n} [1-\frac{{\left(x-x_{i}\right)}^{2}+{\left(y-y_{i}\right)}^{2}}{r^{2}}],$$
where $\hat{f}\left(x,y\right)$ represents the value of kernel density at the spatial position $\left(x,y\right)$, $x_{i},y_{i}$ represents the coordinate of the school i, n represents the number of schools whose distance from position $\left(x,y\right)$ is less than or equal to r, r represents the search radius, x and y represent the coordinates of the center point of the grid to be estimated within the search radius, and ${\left(x-x_{i}\right)}^{2}+{\left(y-y_{i}\right)}^{2}$ represents the square of Euclidean distance between the center point of the grid to be estimated and the school i within the search radius.

### 3.3. Spatial Autocorrelation Analysis

On the basis of classical statistics, spatial autocorrelation analysis is used to test the potential dependence of spatial variables with a certain regularity in different spatial positions [50]. Therefore, this study selected global spatial autocorrelation to explore the average correlation degree, spatial distribution pattern, and significance of PMSs in each unit, used local spatial autocorrelation to detect different spatial aggregation patterns in local spatial regions, and used hot spot analysis method to divide cold and hot spot regions.

Global spatial autocorrelation analysis

We used global spatial autocorrelation analysis to verify the spatial pattern of the whole region and judge whether the phenomenon has aggregation characteristics in space through its internal correlation characteristics [51]. The Moran index (Moran’s I) is generally used for measurement, and the calculation is shown in Equation (4):(4)$$I=\frac{k\sum_{i=1}^{k}\sum_{j=1}^{k}W_{ij}\left(p_{i}-\bar{x}\right)\left(p_{j}-\bar{x}\right)}{\sum_{i=1}^{k}\sum_{j=1}^{k}W_{ij}\sum_{i=1}^{k}{\left(p_{i}-\bar{x}\right)}^{2}},$$
where I represents Moran’s I; $W_{ij}$ represents the spatial weight matrix; $p_{i},p_{j}$ represent the number of schools of research unit i and research unit j, respectively; k represents the number of research units; $\bar{x}$ represents the average of the number of schools of all research units. The weight matrix based on adjacency relation is used in this study, that is, when i and j are adjacent to each other, $W_{ij}=1$, and otherwise it is zero. The value range of Moran’s I is [−1, 1]. The value zero represents uncorrelation, the value greater than zero represents positive correlation, and the value less than zero represents negative correlation.

Local spatial autocorrelation analysis

The local Moran’s I is often used to reveal the spatial correlation and variability between a spatial unit in a region and its neighboring units [52]. This study used the LISA index as a partial form of Moran’s I to test the concentration and dispersion of schools in each unit. The calculation is shown in Equation (5):(5)$$I_{i}=\frac{\left(p_{i}-\bar{x}\right)}{\sum_{i}{\left(p_{i}-\bar{x}\right)}^{2}}\sum_{j}W_{ij}\left(p_{i}-\bar{x}\right),$$

If the local spatial autocorrelation is significant, it means that a certain spatial association or difference exists between the basic unit and its surrounding area. The four kinds of spatial relationships are as follows: “High–High correlation” (H–H) means the high-attribute units are surrounded by high-attribute units; “Low–Low correlation” (L–L) means the low-attribute units are surrounded by low-attribute units; “High–Low correlation” (H–L) represents the low-attribute units surround the high-attribute units; “Low–High correlation” (L–H) represents the high-attribute units surround the low-attribute units [53].

Because local Moran’s I cannot explain whether the cluster is composed of high or low values, we used the Z value of local Getis-Ord $G_{i}^{*}$ hot spot detection to divide the cold and hot spots. A high and positive Z score indicates hot spots. A low and negative Z score represents cold spots [54]. The calculation is shown in Equation (6):(6)$$G_{i}^{*}=\frac{\sum_{j=1}^{k}W_{i,j}p_{j}-\bar{x}\sum_{j=1}^{k}W_{i,j}}{\sqrt[{S}]{\frac{ [k\sum_{j=1}^{k}W_{i,j}^{2}\;-\;{\left(\sum_{j=1}^{k}W_{i,j}\right)}^{2}]}{k-1}}},$$

Meanwhile, the calculation of $\bar{x}$ and S, from Equation (6), is shown in Equations (7) and (8), respectively. Of these, S is the standard deviation.
(7)$$\bar{x}=\frac{\sum_{j=1}^{k}p_{j}}{n},$$
(8)$$S=\sqrt{\frac{\sum_{j=1}^{k}p_{j}^{2}}{k}-{\left(\bar{x}\right)}^{2}}.$$

### 3.4. Index of Dissimilarity

The ID is widely used to measure the degree of difference in the allocation of medical resources [55,56]. The resources of education and health have similarities. We used the relative proportion of education resources of each basic unit and the proportion of the corresponding number of students in a school or the proportion of a geographic area to reflect the overall situation of the fairness of the allocation of education resources in PMSs in the study area. The calculation is shown in Equation (9):(9)$$ID=\frac{1}{2}\times \sum \left|R_{in}-R_{ip}\right|,$$
where i represents each research unit, $R_{in}$ represents the proportion of educational resources of research unit i in the total amount of the study area, including the number of full-time teachers in PMSs and the number of PMSs, and $R_{ip}$ represents the proportion of the number of primary and middle school (PMS) students or geographic area in research unit i to the total number of PMS students or the total geographic area. When $R_{ip}$ refers to the proportion of the number of PMS students in research unit i to the total number of PMS students, the ID can reflect the distribution of educational resources by population (i.e., population distribution). When $R_{ip}$ refers to the proportion of the geographic area in research unit i to the total geographic area, the ID can reflect the distribution of educational resources by geographic (i.e., geographic distribution). The value range of ID is (0, 1). The closer the value to 0, the better the fairness of resource allocation, and vice versa.

### 3.5. Agglomeration Degree

Health resources agglomeration degree refers to the proportion of the number of health resources gathered on 1% of the land area of the province in a city and is often used to evaluate the equity of health resources [57]. In the same way, we constructed an educational resources agglomeration degree (ERAD) to explore the fairness of educational resources in each unit. The calculation is shown in Equation (10):(10)$$ERAD_{i}=\frac{ER_{i}/A_{i}}{ER_{n}/A_{n}},$$
where $ERAD_{i}$ represents the agglomeration degree of educational resources in research unit i, $ER_{i}$ represents the educational resources owned by research unit i, including the number of full-time teachers in PMSs and the number of PMSs, $A_{i}$ represents the land area of research unit i, $A_{n}$ represents the total land area of the upper-level region, and $ER_{n}$ represents the total educational resources of the upper-level region.

When evaluating the equity of educational resource allocation, it is necessary to compare it with population agglomeration degree (PAD). Thus, we used the difference between ERAD and PAD (DEP) to evaluate the equilibrium of educational resources [57]. Of these, the DEP greater than 0 represents the surplus of resources; the DEP less than 0 represents the lack of resources; the DEP equal to 0 represents equilibrium; the PAD reflects the proportion of the population (i.e., the number of PMS students) gathered in a region that accounts for 1% of the land area of the upper-level region. The calculation is shown in Equations (11) and (12):(11)$$PAD_{i}=\frac{Q_{i}/A_{i}}{Q_{n}/A_{n}},$$
(12)$$DEP_{i}=ERAD_{i}-PAD_{i},$$
where $PAD_{i}$ represents the PAD of research unit i, $Q_{n}$ represents the total number of students in the upper-level region, and $Q_{i}$ represents the number of students in research unit i, $DEP_{i}$ represents the equilibrium of educational resources in research unit i.

## 4. Results

### 4.1. Analysis on the Spatial Pattern of PMSs

#### 4.1.1. Spatial Distribution of PMSs

To explore the overall spatial form and difference of PMSs in the study area, we used the SDE to obtain its spatial distribution characteristic ellipse (Figure 3) and related parameters (Table 1). Furthermore, we also explored the spatial form and difference of PMSs in 38 basic units (Table A1 and Figure A1). Among them, the semi-major axis and azimuth angle of the ellipse indicates the direction of the distribution, and the semi-minor axis indicates the range of the distribution. The greater the difference between the semi-major axis and the short axis, the more obvious the directionality of the distribution.

Table 1 and Figure 3 reveal the following: (1) The barycenter of PMSs is located in Anyue County of Ziyang and is also located in the main axis of the Chengdu–Chongqing economic circle. (2) The azimuth of the PMSs is close to 90 degrees, and it tends to be distributed in an east–west direction. (3) From the perspective of the short semi-axis of the ellipse, little difference exists between the PMSs. Relatively speaking, the distribution of primary school is more discrete. (4) The difference between the long half-axis and the short half-axis of the ellipse is higher in middle school than in primary school. The directionality of middle school distribution is more prominent. As shown in Figure A1 and Table A1, there are differences in the spatial distribution characteristics of PMSs between the basic units. However, the spatial distribution characteristics of PMSs in each unit remain broadly consistent (i.e., the parameters of SDE, such as the center coordinate, long and short semi-axes, and direction, differ little between PMSs in each unit).

To further understand the distribution of PMSs in the study area from a micro-perspective, we used the natural split point method to divide the results of KDE into five levels (I-V). Among them, I was the low-density area, II was the relatively low-density area, III was the medium-density area, IV was the relatively high-density area, and V was the high-density area.

From the overall point of view (Figure 4): (1) The PMSs in the Chengdu–Chongqing economic circle have a multicenter spatial structure of “dual cores and multiple sub-centers”, and these cores and centers are distributed in various regions in a cluster. (2) The spatial distribution of primary schools is more balanced than that of middle schools. (3) The spatial distribution of PMSs is consistent with the “one axis, two belts, dual cores, and three districts” in the development plan of the Chengdu–Chongqing economic circle.

In terms of details (Figure 4): (1) The high-density areas of PMSs are distributed in the central area of Chengdu (CAC) and in the MUAC. The relatively high-density areas mainly take the high-density areas as the core and diverge to form a ring or scattered in various areas. (2) The middle and relatively low-density areas expand from the high-density areas or relatively high-density areas, with less sporadic distribution, whereas the low-density areas are distributed along the edge of the study area. (3) The proportion of medium-density and higher-density areas of PMSs is far lower than that of relatively low-density and low-density areas, which reveals the existing problems in the geographic distribution of schools.

#### 4.1.2. Agglomeration Characteristics of PMSs

To analyze the agglomeration of PMSs in the study area from the perspective of districts (counties), we selected global spatial autocorrelation, local spatial autocorrelation, and hot spot analysis methods. From the results of the global spatial autocorrelation (Table 2), Moran’s I of PMSs are 0.28 and 0.24, respectively. They both pass the 1% level of significance test and show a weak positive spatial correlation. This finding indicates that the distribution of PMSs is relatively balanced. Even if the correlation is weak, we can still identify the local agglomeration areas through this method, which will significantly help future planning.

The following are the results of the local spatial autocorrelation analysis (Figure 5): (1) The primary schools include 26 regions with remarkable spatial local autocorrelation. Between them, type H-H is distributed in Nanbu County, Yilong County, and Santai County, as well as in the MUAC and its surrounding areas (e.g., Banan District, Hechuan District). Type H-L is located in Yibin County. Type L-H is distributed in the city of Huaying and Dadukou District. Type L-L is distributed in Meishan (e.g., Danling County), Leshan (e.g., Shawan District), and Ya’an (e.g., Hanyuan County). (2) The middle schools include 20 regions with remarkable spatial local autocorrelation. Between them, type H-H is distributed in the surrounding areas of the city of Jianyang and Yubei District. Type L-L is distributed in Pengshui autonomous county, Shizhu autonomous county, and the southwest corner of the Chengdu–Chongqing economic circle. Type L-H is located in Pengshan County, Qingshen County, and Dadukou District.

The results of the hot spot analysis (Figure 6) reveal a big difference in the hot spot distribution of middle schools and primary schools. The hot spots of primary schools are concentrated in the MUAC, Dawan Town concentrated area, and Nansuiguang Town concentrated area. The hot spots of middle schools are concentrated mainly in the surrounding areas of Jianyang. The cold spots of PMSs are distributed mainly in the surrounding districts (counties) of Jinkouhe District.

### 4.2. Analysis on the Fairness of Educational Resources

#### 4.2.1. Temporal Variation Characteristics of Fairness on a Global Scale

According to the number of full-time teachers and PMSs, we calculated the ID of educational resources in PMSs in the study area from 2010 to 2019. As shown in Figure 7, the ID of educational resources in PMSs ranges from 0.02 to 0.21, indicating that the overall allocation of resources is relatively fair. The ID according to the population distribution shows a downward trend, whereas the ID according to the geographic distribution does not change significantly. The distribution according to the population is better than the distribution according to the geographic location.

The ID of educational resources in middle schools is between 0.04 and 0.23. In addition to the declining trend in the ID of full-time middle school teachers according to geographic distribution, the others—including the ID of full-time middle school teachers according to population distribution, the ID of middle schools according to geographic distribution, and the ID of middle schools according to population distribution—show an upward trend. The ID of educational resources in primary schools is between 0.03 and 0.28. Except for the fluctuation in the ID of full-time primary school teachers according to geographic distribution, the others—including the ID of full-time primary school teachers according to population distribution, the ID of primary schools according to geographic distribution, and the ID of primary schools according to population distribution—show a downward trend. We discovered that the distribution of educational resources in primary schools is more balanced than that in middle schools, and this distribution shows a good development trend.

#### 4.2.2. Temporal Variation Characteristics of Fairness on a Local Scale

This study calculated the ERAD of each unit from 2010 to 2019. Because the values and trends of ERAD were similar in each year, we selected 2010, 2014, and 2019 as a representative time sample to display. As shown in Figure 8, significant differences in ERAD exist between each unit. Specifically, full-time teachers are relatively concentrated in the main urban areas of Chengdu and Chongqing, but they are relatively scarce in Ya’an. Middle schools are distributed mainly in Chengdu, Nanchong, Guang’an, and Ziyang, and the main urban areas of Chongqing, whereas they are relatively sparse in Ya’an and Pengshui County. As for the agglomeration degree of primary schools and PMSs, Yongchuan District, Dazu District, and Rongchang District are the main gathering places, and Ya’an is a sparsely populated place.

To evaluate the fairness of the distribution of educational resources, we used the DEP as a measurement standard (Figure 9). To facilitate analysis, we divided the changes in regional equilibrium from 2010 to 2019 into small fluctuation type, positive growth type, and negative growth type.

In Figure 9a (equilibrium of full-time primary school teacher), the negative growth type is identified in Leshan, Deyang, Yongchuan District, and Dazu District. Among them, Leshan and Deyang tend to develop in a balanced manner, whereas Dazu District and Yongchuan District have transitioned from types of resource surplus to resource scarcity. The positive growth type is mainly found in Dazhou, Guang’an, and Dianjiang County, all of which have turned from types of scarce resources to excess resources. The small fluctuation type is found mainly in Meishan, Ya’an, and Qianjiang District, which basically have achieved equilibrium. In Figure 9b (equilibrium of the number of primary schools), the positive growth type is found mainly in Rongchang District, Dazu District, and Tongnan District, which are in a state of excess resources and maintain growth. The negative growth type is found mainly in Chengdu, the MUAC, Zigong, Ziyang, and Yongchuan District. Except for the shortage of resources in the MUAC and Chengdu, other regions tend to develop in a balanced manner. The smaller fluctuation type is found mainly in Mianyang, Meishan, Fuling District, and Tongliang District, which have also maintained a long-term equilibrium state. In Figure 9c (equilibrium of full-time middle school teacher), the negative growth type is found mainly in Ziyang, the MUAC, and Dazu District, all of which have shifted from excess resources to a balanced state. The positive growth type is found mainly in Guang’an, Hechuan District, and Wanzhou District, all of which have transformed from scarce resources to a balanced state. The small fluctuation type is found mainly in Luzhou, Changshou District, and Nanchuan District, which have both surplus and scarce resources. In Figure 9d (equilibrium of the number of middle schools), the negative growth type is found mainly in Ziyang, Rongchang District, and the MUAC. Among them, Ziyang develops in a balanced manner, and other areas are experiencing a lack of resources. The positive growth type is found mainly in Chengdu, Guang’an, and Nanchong, all of which are experiencing a state of surplus of resources. The small fluctuation type is found mainly in Dazhou, Kaizhou District, and Zhongxian County, which have both surplus and scarce resources. In Figure 9e (equilibrium of full-time teachers in PMSs), the positive growth type is distributed mainly in Dazhou, Guang’an, and Tongnan District. The negative growth type is distributed mainly in Dazu District, the MUAC, and Yongchuan District. The small fluctuation type is distributed mainly in Luzhou, Ya’an, and Qijiang District. In Figure 9f (equilibrium of the number of PMSs), the positive growth type is distributed mainly in Meishan, Suining, and Changshou District. The negative growth type is distributed mainly in Yibin, Zigong, and Yongchuan District. The small fluctuation type is distributed mainly in Luzhou, Fuling District, and Nanchuan District.

Figure 8 shows that Chengdu, as one of the dual-core regions of the Chengdu–Chongqing economic circle, is not as prominent as the MUAC. According to the results of KDE, however, the level of kernel density of Chengdu’s downtown area should be comparable to that of Chongqing’s main urban area. Therefore, this study made a detailed division of the administrative divisions of Chengdu and categorizes it into 10 research units. It specifically included the CAC (the CAC includes Wuhou District, Qingyang District, Longquanyi District, Jinjiang District, Wenjiang District, Chenghua District, Jinniu District, Shuangliu District, Pidu District, Xindu District, Qingbaijiang District) and nine surrounding jurisdictions (Jianyang, Dujiangyan, Pengzhou, Qionglai, Chongzhou, Jintang County, Dayi County, Pujiang County, Xinjin District). Because of the lack of data, this study counted only the ID (Figure 10), ERAD (Figure 11), and equilibrium (Figure 12) of the educational resources in PMSs from 2017 to 2019.

According to Figure 10, Figure 11 and Figure 12, the ID of educational resources from 2017–2019 is basically consistent with the previous results in terms of value and trend. It can be seen, however, that the ERAD in the CAC is much higher than that in the main area of Chongqing. The changing trends in other regions are consistent with the previous trends, which shows that a subdivision of Chengdu is necessary. From the perspective of equilibrium, the changes in the remaining units are not obvious, except for a few units that have large changes, so a longer time series is needed to reflect the trend of changes in the equilibrium of educational resources.

## 5. Discussion

### 5.1. Advantages of Proposed Method over Conventional Methods

Taking the Chengdu–Chongqing economic circle, which is the fourth-largest growth center of China, as the study area, we measured the spatial pattern and equity of educational resources by methods of KDE, SDE, spatial autocorrelation analysis, ID, and agglomeration degree. Compared with traditional studies, such as the standard deviation [16,17] and Gini coefficient [1,3,18,19] of years of schooling among students, this study method has more advantages in visualization and structure (e.g., the distribution of educational resources can be clearly seen through KDE and SDE). Next, compared with spatial analysis studies, such as statistical maps [39], buffer analysis [41], and Ripley’s K function [40], this study method revealed the temporal variation characteristics of the equity of educational resources (e.g., ID and agglomeration degree for long time series). In summary, we combined the advantages of traditional methods and spatial analysis methods to effectively reveal the spatial pattern and equity of educational resources. By the end of 2020, the county compulsory education in China reached the goal of basically balanced development, which provided strong support for the conclusion of this study. Furthermore, POI data used in this study can overcome the shortcomings of traditional data (i.e., the statistics may be incomplete or difficult to obtain), and it has been widely used in the fields of traffic, crime, health, and urban development [58,59,60,61,62].

### 5.2. Factors Affecting the Fairness of Educational Resources

Presently, numerous studies have shown that socio-economic development, population distribution, education policies, traffic network, and the natural environment affect the equity of educational resources. Since the implementation of the Compulsory Education Law, the level of basic education and the quality of teaching in China have improved significantly. However, uneven resource allocation remains an obstacle to achieving equity in education (consistent with the results reflected by ID and agglomeration degree in this study). Of these, teacher–student ratio and education funding are important influencing factors [63]. The gap between urban and rural areas also profoundly affects the equity of educational resources. Specifically, urban areas (e.g., the CAC and MUAC), with a dense population, well-developed transport network, and rapid economic development, have an abundance of educational resources [42,64,65]. In rural areas, the opposite is true. In addition, the natural environment, as a fundamental condition that forms the spatial pattern of educational resources, can directly influence the spatial equilibrium of educational resources (e.g., topography and river systems) [66].

### 5.3. Limitations and Future Research Directions

This research could be improved in the following aspects. First, this study uses the POI data only for 2020, which fails to present the changes in the spatial pattern of educational resources over a long period of time. Second, in the analysis of spatial pattern, it is also possible to explore the spatial agglomeration characteristics of educational resources at different distance scales based on Ripley’s K function. Third, the fairness of educational resources is affected by many factors, and this study considers only the number of schools and full-time teachers. In future studies, the regional economic level and policies can be included. Finally, we recommend conducting an in-depth analysis of how factors affect the equity of educational resources.

## 6. Conclusions

This study combined the advantages of big data and geography to depict the spatial distribution and agglomeration characteristics of PMSs. At the same time, the ID and agglomeration degree was used to measure the fairness of educational resources in the two dimensions of geography and population, thereby filling the research gaps in the region. The results show the following: (1) From the perspective of spatial distribution, the PMSs are in a multicenter spatial structure of “dual cores and multiple sub-centers”, which is consistent with the development plan of “one axis, two belts, dual cores, and three districts” in the Chengdu–Chongqing economic circle. Through the SDE parameters, the PMSs tend to be distributed in the east–west direction, and the barycenter is located in Anyue County, Ziyang. Primary schools are more discrete in spatial distribution, whereas middle schools are distributed in a more obvious direction. (2) Moran’s I of PMSs are 0.24 and 0.28, respectively, and both pass the 1% level of the significance test. This result shows a weak positive spatial correlation and indicates that the distribution of PMSs is relatively balanced. The agglomeration areas of PMSs, however, are quite different. (3) On the whole, the ID of educational resources in PMSs from 2010 to 2019 is between 0.02 and 0.21, which shows that the allocation of resources is relatively balanced, and the distribution by population is better than that by geography. Separately, the development of educational resources in primary schools is more balanced than that in middle schools. Regarding internal units, results show obvious differences in the agglomeration degree and equilibrium of educational resources. Policy, economic level, geographical location, natural environment, etc., may be the main reasons for unfairness in education resources.

Through exploring the spatial pattern and fairness of educational resources in PMSs, this study makes the following recommendations for future planning. In low-density and relatively low-density areas (Figure 4), online teaching and building new schools are effective ways to promote equity in education. In Figure 9 and Figure 12, increasing the number of enrolments (the difference between ERAD and PAD is greater than 0) and teaching posts (the difference between ERAD and PAD is less than 0) is beneficial to the rational allocation of educational resources. Moreover, economic development and policy making will contribute to achieving equity in education.

## Figures and Tables

**Figure 1 ijerph-19-10840-f001:**
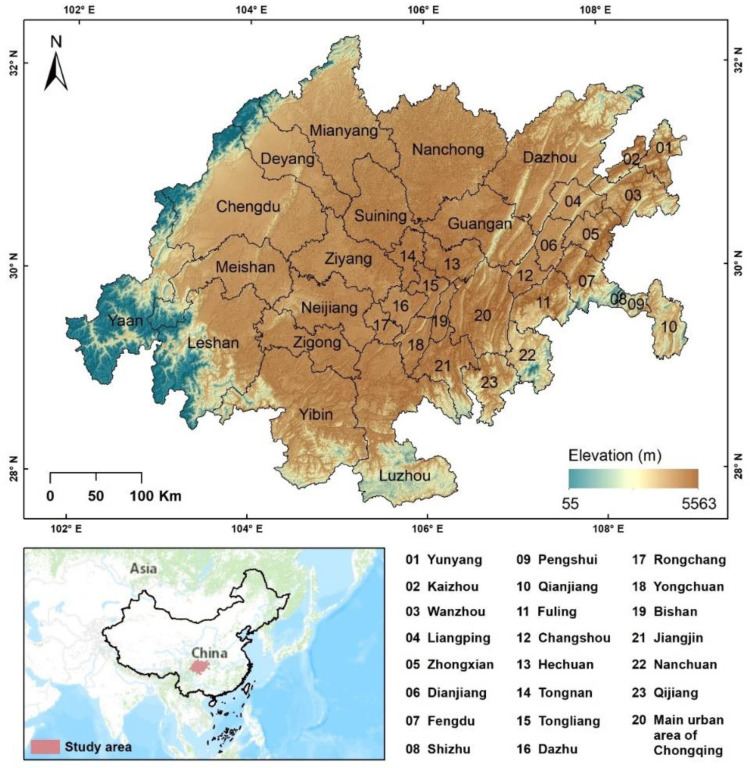
Location of the study area and its 38 basic units.

**Figure 2 ijerph-19-10840-f002:**
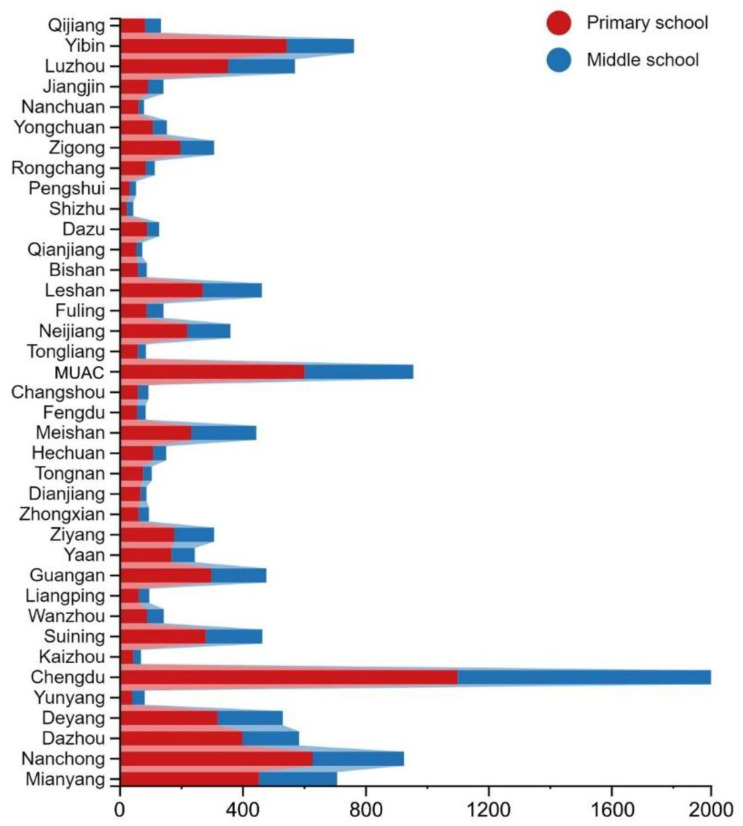
Number of PMSs in each study unit.

**Figure 3 ijerph-19-10840-f003:**
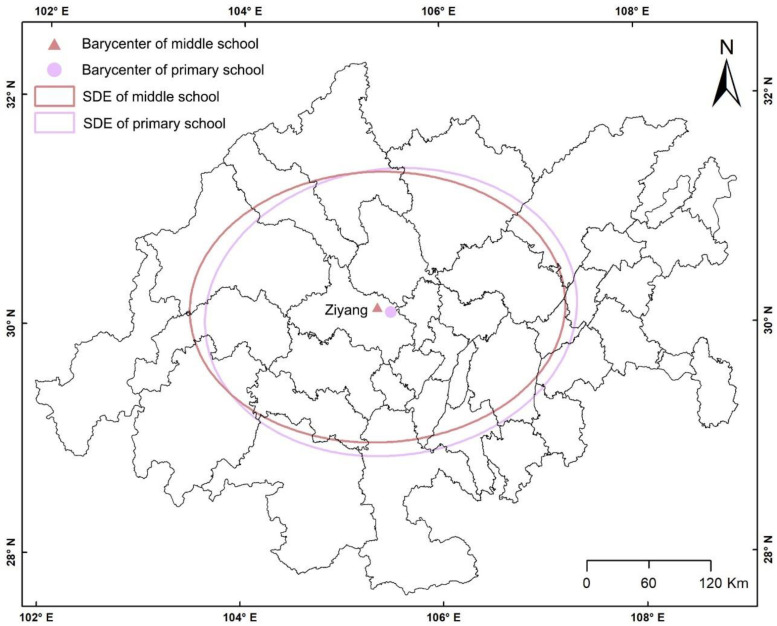
The SDE of PMSs.

**Figure 4 ijerph-19-10840-f004:**
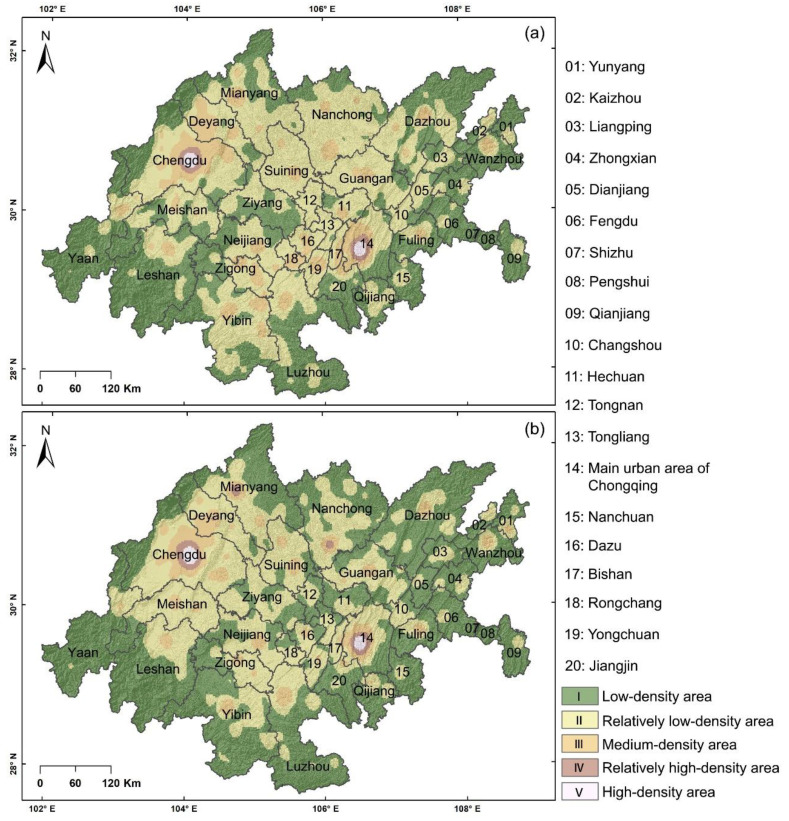
(**a**) Results of KDE of primary schools, and (**b**) results of KDE of middle schools.

**Figure 5 ijerph-19-10840-f005:**
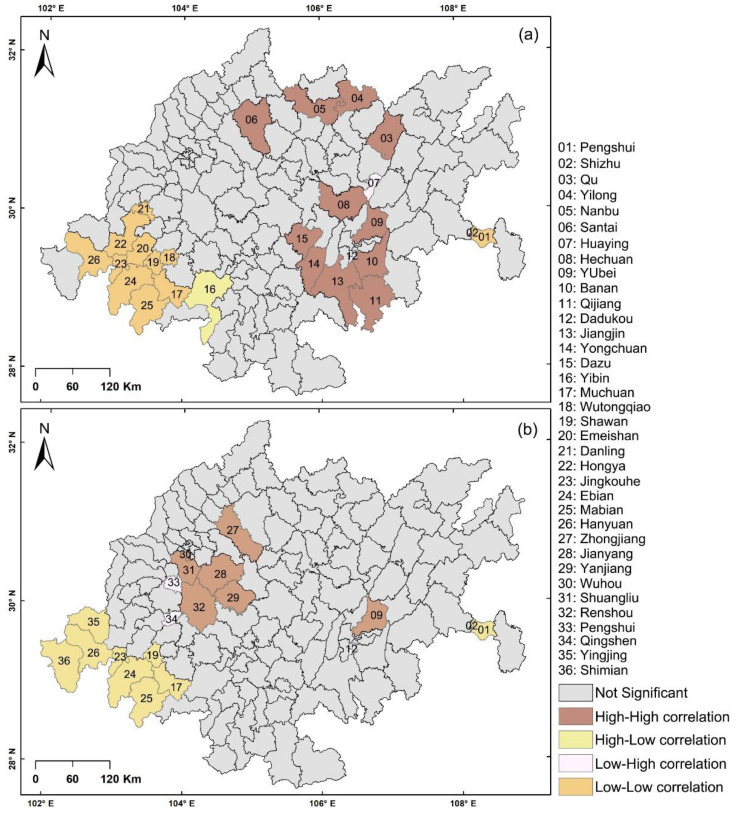
(**a**) Results of local spatial autocorrelation analysis of primary schools, and (**b**) results of local spatial autocorrelation analysis of middle schools.

**Figure 6 ijerph-19-10840-f006:**
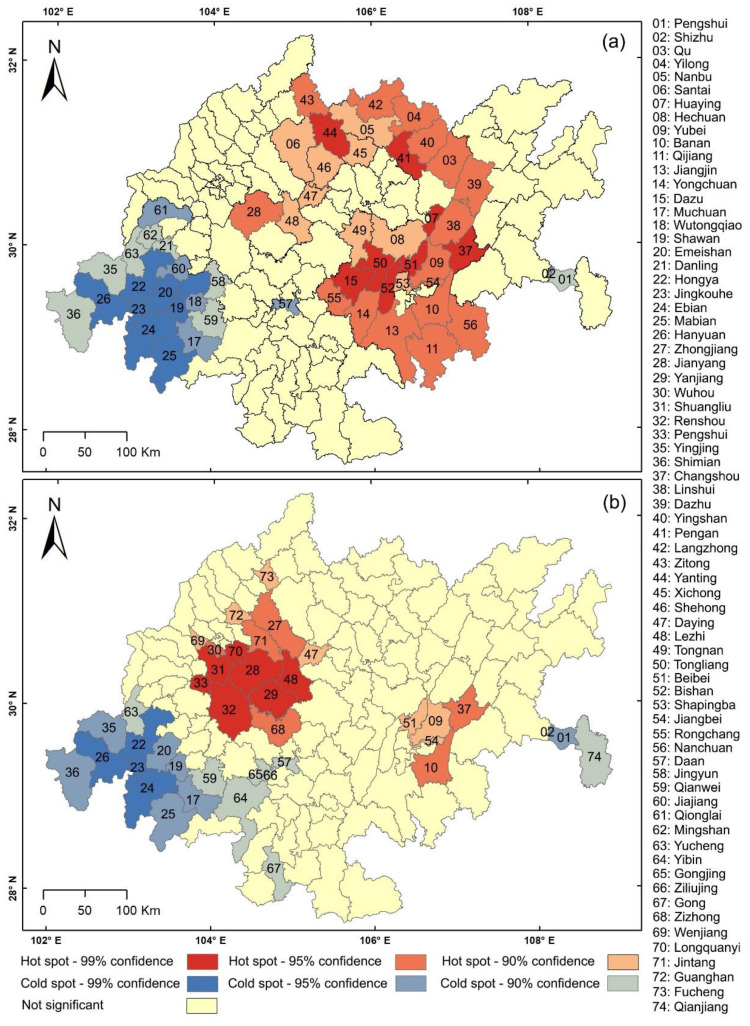
(**a**) Results of hot spot analysis of primary schools, and (**b**) results of hot spot of middle schools.

**Figure 7 ijerph-19-10840-f007:**
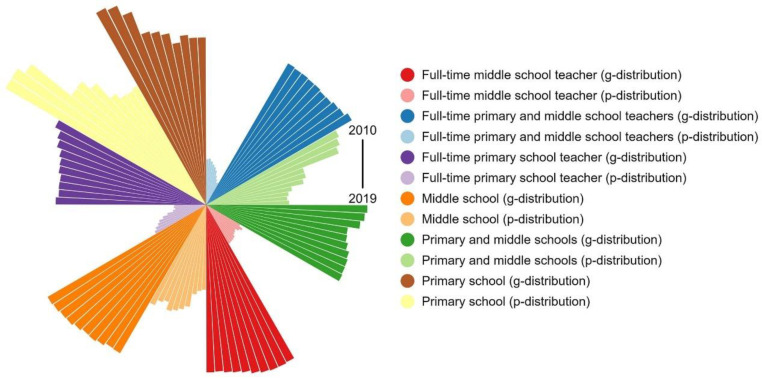
ID from 2010 to 2019. P-distribution represents population distribution; g-distribution represents geographic distribution; time increases in the clockwise direction.

**Figure 8 ijerph-19-10840-f008:**
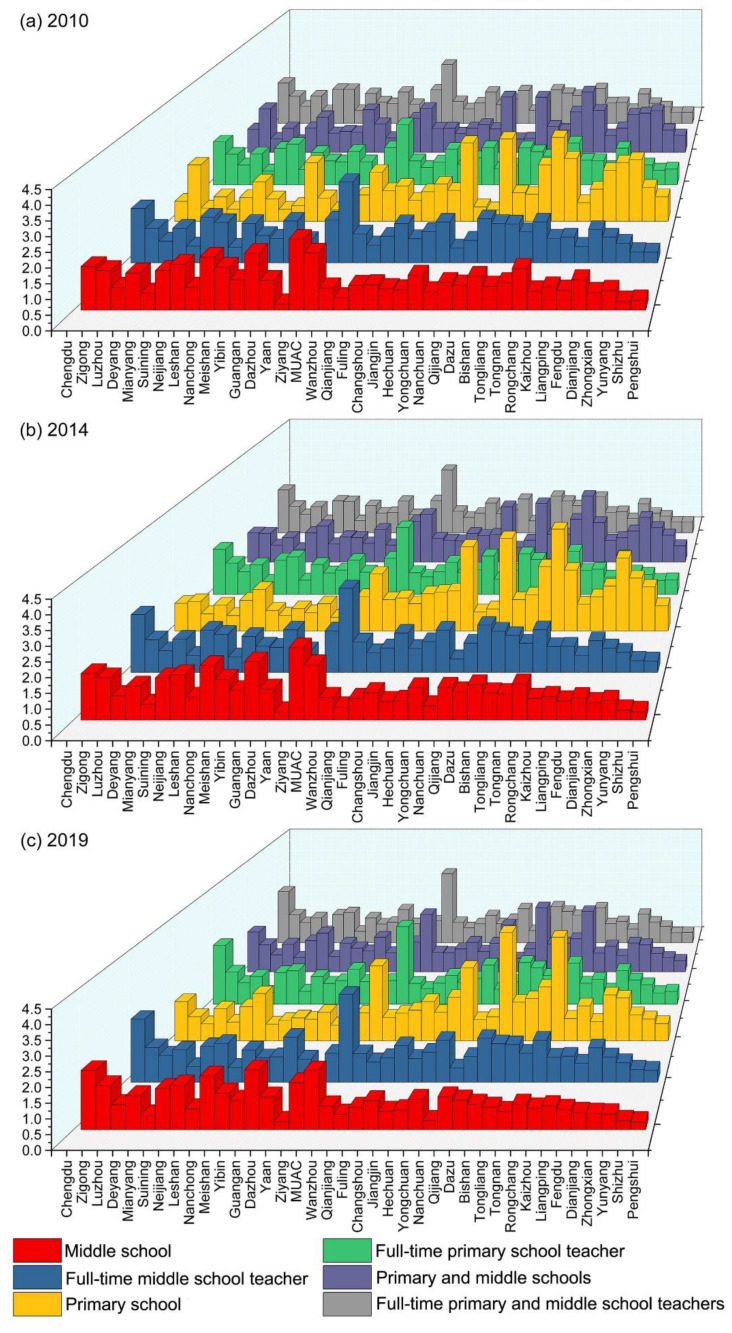
ERAD in (**a**) 2010; (**b**) 2014; and (**c**) 2019.

**Figure 9 ijerph-19-10840-f009:**
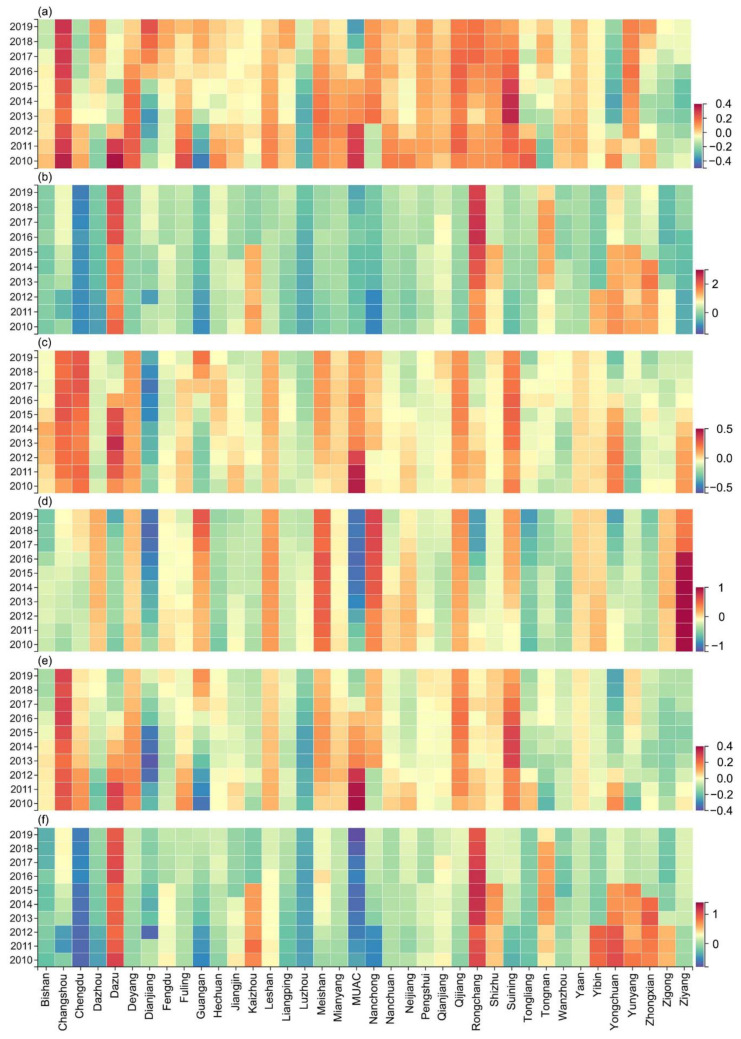
DEP from 2010 to 2019. (**a**–**f**), respectively, represent the changes of equilibrium in full-time primary school teachers, the number of primary schools, full-time middle school teachers, the number of middle schools, full-time PMS teachers, and the number of PMSs.

**Figure 10 ijerph-19-10840-f010:**
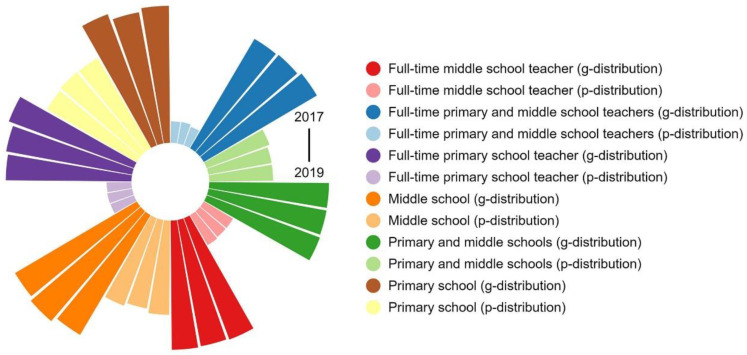
ID from 2017 to 2019. p-distribution represents population distribution; g-distribution represents geographic distribution; time increases in the clockwise direction.

**Figure 11 ijerph-19-10840-f011:**
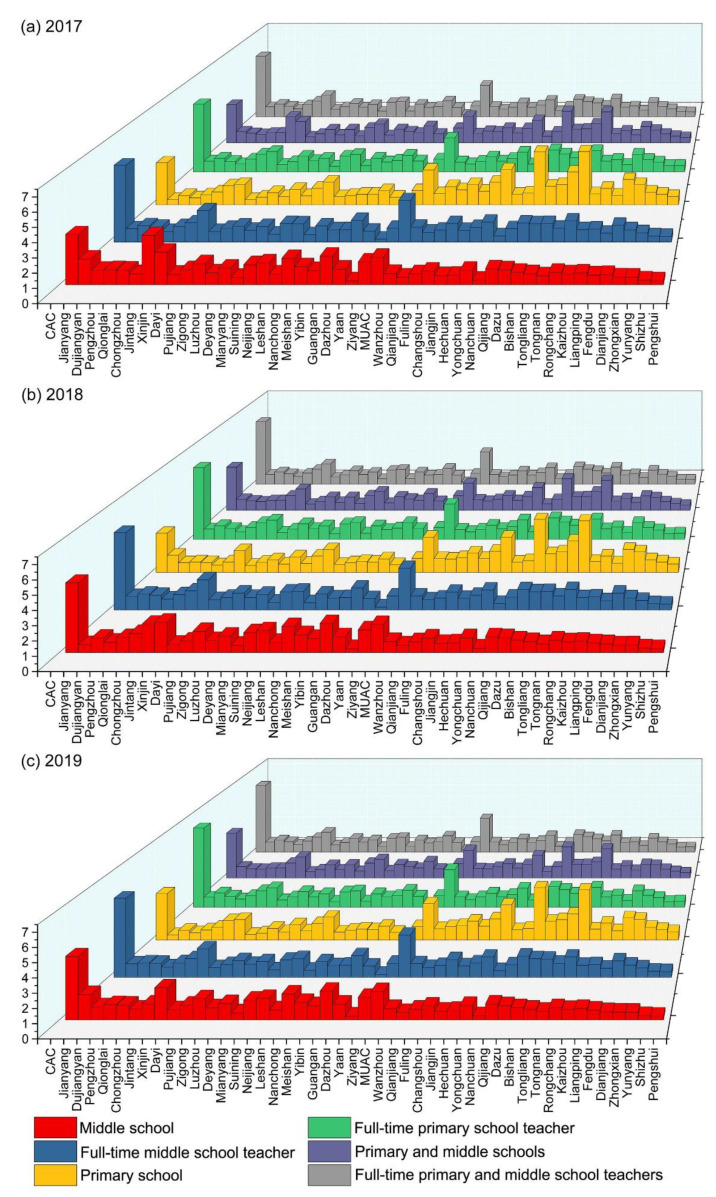
ERAD from 2017 to 2019.

**Figure 12 ijerph-19-10840-f012:**
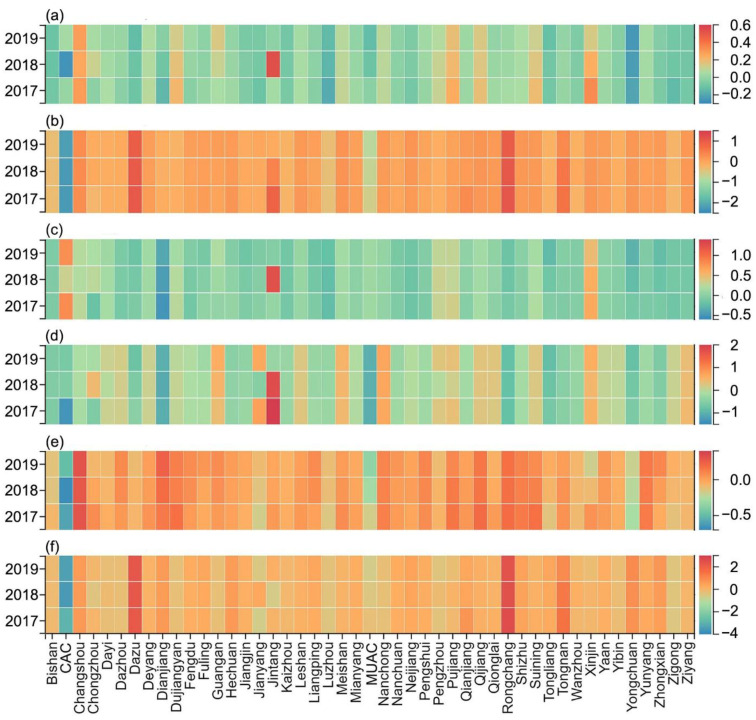
DEP from 2017 to 2019. (**a**–**f**), respectively, represent the changes of equilibrium in full-time PMS teachers, the number of PMSs, full-time middle school teachers, the number of middle schools, full-time primary school teachers, the number of primary schools.

**Table 1 ijerph-19-10840-t001:** The parameters of SDE.

Type	Center-x	Center-y	StdDist-x	StdDist-y	Rotation
Primary school	105.49	30.14	1.87	1.25	85.20
Middle school	105.36	30.18	1.89	1.18	88.64

Note: Center-x and Center-y represent the center coordinate of the SDE; StdDist-x and StdDist-y represent the long and short semi-axes of the SDE, respectively; rotation represents the direction of the distribution of schools (the due north direction is 0°); unit of these parameters is degree.

**Table 2 ijerph-19-10840-t002:** The results of global spatial autocorrelation.

Type	Moran’s I	Z-Score	*p*-Value
Primary school	0.28	5.64	0
Middle school	0.24	4.85	0.000001

Note: Moran’s I stands for correlation, and its value ranges from −1 to +1. A value of Moran’s I greater than 0 represents a positive correlation. The larger the value, the stronger the spatial correlation. A value of Moran’s I less than 0 represents a negative correlation. The smaller the value, the greater the spatial difference. If Moran’s I equals zero, the spatial objects are randomly distributed. Z-score means a multiple of the standard deviation, which reflects the degree of dispersion between individuals within the group. *p*-value means the possibility that the observed spatial pattern is created by a stochastic process.

## Data Availability

Not applicable.

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
