# Peer review of "Spatial Pattern and Fairness Measurement of Educational Resources in Primary and Middle Schools: A Case Study of Chengdu–Chongqing Economic Circle"

_ijerph, 2022, doi:10.3390/ijerph191710840_

Round 1
Reviewer 1 Report
Dear Sirs,
It was a pleasure to review your research. Some corrections given in a document are necessary to make the results understandable to a wider audience.
Best regards

Reviewer 2 Report
This paper is relatively well written but the core science is not in place yet.
1. The term "educational resources" is used in the title and throughout. This is not explained anywhere. Do you refer to schools, teachers, students? This must be clarified. Also, why is this of interest? This makes the science of the paper unclear.
2. ln 69: This sentence doesn't make any sense?
3. ln 74: Fix the language
4. In Figure 1 the basic units are numbered and in a certain order. This order is not maintained through the paper, making following of the results very difficult.
5. Page 3: make the web links footnotes rather.
6. Page 3: What is a PMS? It is never defined or explained.
7. lns 112 - 114: Explain for the reader the age structure of students in Chinese schools.
8. ln 118: What does this sentence mean to say?
9. ln 130: Why capital X and Y? Rathe small x and y and be consistent into the rest of the sections.
10. ln 131: ..." is the total..."
11. Pages 5 - 7: There shouldn't be indented space after every line after an equation.
12. ln 159: What is n now? Was m?
13. ln 183: n again but different terminology used?
14. ln 207-208: not at all clear.
15. eq 7+8: x not X
16. ln 214: "the population refers to the students" - how? Not clear what is actually being analysed.
17. The Results and Discussion sections both have results and discussions. These should be separated out better. Results only in results and only discussion points and observations in the discussion section.
18. Figure 3: What about ellipses per basic unit? That would be much more informative.
19. Figure 4 caption: fix the language
20. Section 4.2: Why now the number of teachers? This could also have been investigated using the methodology in 4.1? The aims of all the analysis is not clear.
21. Page 13: g-distribution and p-distribution come out of no where? Explain?
22. Figure 7 is not referred to in the text.
23. ln 334: "to display" instead of "for analysis". You did all the analysis?
24. Figure 8 (and the similar one later) are inappropriate. The basic units are categorical and can't be plotted continuously as here. If you change the order of the areas, everything changes. Reconsider everything here. There are appropriate spatio-temporal techniques to use.
25. Figure 9: labels for the units? Very unclear what the idea of this graphic is.
Overall, how has fairness been determined? There is no comparison to other characteristics of the basic units such as population density or accessibility.
Reviewer 3 Report
The abstract and keywords are formulated properly. The article is structured correctly. However, in the methodological chapter, one might ask whether a third level of subsections (3.3.1., etc.) is really needed. Perhaps it would be more justified to use a different form of division, e.g. bullet points, thus avoiding adding further subchapters to the already extensive structure of the article.
In my opinion, websites should also be cited with consecutive numbers, and the full link should only be included in the literature list (e.g. line 109 and others). In addition, there are no spaces in parentheses after commas.The graphics are correct. However, on figure 4 it would be necessary to explain what the numbers I-V mean (there is room in the legend for adding such information) so that the map is legible without the need to read the text at the same time. The scale should be placed horizontally in relation to the map layout (it can be placed directly on the contents of both maps). The same remarks apply to figure 5. Legends on maps and charts should be horizontal – error in e.g. figure 8.
While the subject of the article is important, especially socially, the paper does not explore the broader international influence and context of this topic. Internationalization of own research results is sorely lacking, as the journal has a wide international reach. Therefore, the literature list should be expanded to include English-language items, and thus the literature review section requires expansion as well. The “Discussion" chapter also requires a major remodeling. The discussion should be a polemic of author’s own achievements with international research. The discussion should include only the figures that disrupt the existing methods, models, theories, and contribute something new to global discourse – as there should be no result figures in this chapter, they should be moved to e.g. the empirical chapter. The narrative should incorporate the current international state of knowledge on this subject into own results or methods.
Round 2
Reviewer 1 Report
Dear authors,
Thank you for accepting the reviewer's suggestions.
Author Response
Great thanks for the time and effort you expend on this paper.
Reviewer 2 Report
Thank you for the revised paper. However, a number of fundamental issues still need attention. I refer to the original review points:
Point 1: "Education Resources" should be defined earlier on, in the abstract and introduction. Otherwise, a reader cannot follow until Section 3.
Point 3: The sentence you have now added lns 93 - 94 is the strong point. However, it is not evident for me how you actually measure fairness. The techniques used are spatial distribution type techniques, but how does that precisely translate to fairness? I feel a jump is being made without literature backing it nor new methodology being proposed. Lns 138-146 again state only the methods used, but no motivation how these relate to fairness?
Point 4: But Figure 5 and 6 have the same basic units and the numbering is different?
Point 6: PMS is only defined at lines 240-241, however it is already used earlier. It should be defined the first time it is mentioned.
Point 14: The same notation for different things? This needs to be cleaned up.
Point 18: How is the conclusion "the spatial distribution characteristics of PMSs in each unit remain largely consistent" made? More information should be added here.
Point 21: g- and p-distribution are still not defined. They just appear as notation.
Figure 8 and 9 should similar results but the captions are structured differently. Some consistency should be added here.
Point 25: I think a citation is needed here and a better explanation on what is being shown, how it is calculated (exactly with a formula) and how it should be interpreted. This may be the link to defining how to jump to measuring fairness. Some time should be spent on this.
Reviewer 3 Report
No comments for authors
Author Response

(The authors gave the same response as above.)
